# Association of Circulating Endothelial Nitric Oxide Synthase Levels with Phosphataemia in Patients on Haemodialysis

**DOI:** 10.3390/biomedicines12030687

**Published:** 2024-03-19

**Authors:** Leszek Niepolski, Kamila Malinowska-Loba

**Affiliations:** Faculty of Medicine, Poznań Academy of Medical Sciences, Bułgarska Str. 55, 60-320 Poznan, Poland; malinowska.kamila.maria@gmail.com

**Keywords:** end-stage renal disease, eNOS, haemodialysis, phosphataemia

## Abstract

The amount of evidence indicates that hyperphosphataemia (HP) can induce endothelial damage and significantly impair endothelial nitric oxide synthase (eNOS) expression. There are no clinical studies that have assessed HP and its correlation with circulating eNOS concentration in patients with end-stage renal disease (ESRD). Our preliminary study aimed to evaluate the relationship between plasma inorganic phosphorus (P) levels and circulating plasma eNOS concentration in patients on haemodialysis (HD). A total of 50 patients on HD were enrolled to the study. They were divided into groups according to the tertiles of P. The examined HD group was also analysed and compared with controls as a whole group; then, the group was divided into patients with and without dyslipidaemia (D) as well as into those with and without type 2 diabetes mellitus (type 2 DM). A total of 26 age-matched healthy volunteers were included in the study as the control group. The plasma levels of eNOS in HD patients are reduced in comparison to those in healthy subjects. There was no difference in plasma eNOS concentrations between HD patients with type 2 DM and those without DM as well as between those with D and without D. In the entire group of HD patients, there were positive correlations between circulating levels of eNOS and plasma P concentrations. In HD patients with D, higher systolic and diastolic blood pressure were accompanied by decreased plasma eNOS concentrations. In conclusion, HP and high blood pressure appear to decrease the circulating eNOS levels. These findings demonstrate an additional negative impact of HP on eNOS activity.

## 1. Introduction

The amount of evidence indicating that endothelial dysfunction is a characteristic of end-stage renal disease (ESRD) is increasing [1,2,3]. Compared to healthy individuals, ESRD patients have a much higher risk of more severe endothelial failure; one of the biochemical markers of endothelial failure is hyperphosphataemia (HP) [4,5]. HP is considered to be one of the most important uraemic toxins and is an independent risk factor for accelerated cardiovascular disease in ESRD [4,6]. Emerging evidence shows that HP, both in vitro and in vivo, can induce endothelial damage [4,5]. It accelerates along with a decrease in the glomerular filtration rate and is known to reduce nitric oxide (NO) bioavailability [4]. NO is synthesised in the endothelium by the constitutive endothelial nitric oxide synthase (eNOS) enzyme [4]. Recent studies using cell culture models have indicated that the eNOS activity can be inhibited by several uraemic toxins, such as oxidative stress, inflammation, hyperhomocysteinemia, and HP; these toxins can accumulate in plasma during ESRD [4,5,6]. It is very difficult to determine which of the uraemic toxins has the most significant impact on the reduction in eNOS activity. Recently, researchers have focused on HP as a uraemic toxin that significantly impairs eNOS expression in endothelial cells (ECs) [5,7,8]. However, there are no clinical studies that have assessed HP and its correlation with the circulating eNOS concentration in patients with ESRD. Our preliminary study aimed to evaluate the relationship between plasma inorganic phosphorus (P) levels and the circulating plasma eNOS concentration in patients on haemodialysis (HD).

## 2. Materials and Methods

### 2.1. Patients on HD

All patients lived in the Greater Poland Voivodship, Poland, and were dialyzed in B.Braun Avitum Poland, Nowy Tomyśl Dialysis Center. A total of 50 patients on HD (27 men and 23 women; aged 66 (60–74) years; HD vintage, 29 (15.1–47.4) months) were treated with maintenance HD for a minimum of 6 months. All the patients underwent HD three times for 4 h per week on low-flux polysulphone-based membranes with a surface area of 1.4–2.1 m^2^. The dialysers were not reutilised. Blood flow was in the range of 180–370 mL/min, and dialysate flow was in the range of 500–650 mL/min. Low-molecular-weight heparin was used as an anticoagulant. The duration of treatment with HD was defined as the number of months from the commencement of the first HD session that started regular HD treatment to enrolment in the present study.

The adequacy of the HD treatment was assessed for each patient by calculating the average urea dialysis clearance normalised to the urea distribution volume (Kt/V_urea_) from the monthly measurements that were taken 3 months prior to the study. The urea kinetic model of Daugirdas [9] was used to determine the second-generation logarithmic estimates of the single-pool variable volume Kt/V_urea_. The patients on HD who fell into one or more of the following categories were excluded: (i) neoplastic disease, active collagen disease, acute coronary syndrome, or cerebral stroke in the 3 months preceding the commencement of the study; (ii) a constant dose of oral phosphate binders for at least 1 month; and (iii) parathyroidectomy or treatment with oral calcimimetic agent cinacalcet in the 12 months preceding the commencement of the study. None of the patients were receiving corticosteroids or immunosuppressant drugs at the time of the study.

The total erythropoiesis-stimulating agent (ESA) dose prescribed for 1 month was averaged for a week, and the resulting mean in IU/kg per week was used. The darbepoetin alfa dose was converted to the ESA dose (IU/kg per week) by multiplying the darbepoetin alfa dose prescribed for 1 week (µg) by 200.

### 2.2. Controls

A total of 26 age-matched healthy volunteers from Greater Poland Voivodship were included in the study as the control group. No substantial deviations were recorded during the medical interview and physical examination. The volunteers had not received phosphate binders in the 12 months before the commencement of the study.

### 2.3. Study Design

The HD patients were divided into groups according to the tertiles of P (mg/dL): P ≤ 3.4, P = 3.5–5.5, and P ≥ 5.6. The groups were compared to each other and analysed separately. Due to the inconsistent scientific reports about the influence of lipid and glucose metabolism on plasma eNOS [10,11,12] levels, the examined HD group was also analysed and compared with the controls as a whole group; then, the group was divided into patients with dyslipidaemia (D) and patients without D, as well as into those with type 2 diabetes mellitus (type 2 DM) and those without DM. The participants were categorised as having D when they met the criteria of the K/DOQI guidelines [13]. The entire HD group (n = 50) included 25 type 2 DM patients. There were no patients with type 1 DM. There were no subjects with DM in the control group. The scheme of the study is shown in Figure 1.

### 2.4. Ethics Approval of Research

The study was performed in accordance with the principles of the Declaration of Helsinki [14] and approved by the Institutional Review Board of Poznań University of Medical Sciences, Poland. Written informed consent was obtained from all subjects before participation.

### 2.5. Clinical and Laboratory Methods

Blood pressure was measured before the HD session by certified staff using standardised procedures. It was measured in the right arm with a sphygmomanometer after the participant was seated and had rested for 5 min; two measurements, taken 30 s apart, were recorded. Pulse pressure (PP) was defined as the difference between systolic blood pressure (BP_sys_) and diastolic blood pressure (BP_dias_). Mean arterial pressure (MAP) was determined as BP_dias_ + 1/3(BP_sys_ − BP_dias_).

In the HD patients, dry body weight (DBW) was determined according to the definition given by Kouw et al. [15] as the weight at the end of an HD session, when a patient is often likely to develop symptoms of hypotension. In other words, DBW is the lowest weight a patient can tolerate without the development of symptoms of hypotension. DBW was determined by an experienced nephrologist. Body mass index BMI (kg/m^2^) was calculated as DBW (kg)/height (m)^2^.

Complete blood count was assessed using an automated haematology analyser manufactured by Sysmex XN-1000 (Kobe, Japan). From the obtained parameters, the concentration of haemoglobin (HgB) and the number of red blood cells (RBCs), white blood cells (WBCs), platelets (PLTs), and haematocrit (Ht) were analysed.

Dyslipidaemia in the control group was assessed according to the European Society of Cardiology and the European Atherosclerosis Society (ESC/EAS) guidelines [16]. Dyslipidaemia in HD patients was diagnosed according to the Kidney Disease Outcomes Quality Initiative (K/DOQI) Clinical Practice Guidelines for Managing Dyslipidaemias in Chronic Kidney Disease (2003) [13]. The patients diagnosed with dyslipidaemia after showing serum low-density lipoprotein cholesterol (LDL-chol) ≥ 100 mg/dL were referred to as hyper-LDL cholesterolaemic patients, whereas those showing non-high-density lipoprotein cholesterol (non-HDL chol) ≥ 130 mg/dL and triglycerides (TG) ≥ 200 mg/dL were described as hyper-TG/hyper-non-HDL cholesterolaemic patients. The patients who met one of these criteria were included in the dyslipidaemic group. The remaining patients were referred to as non-dyslipidaemic, in accordance with the K/DOQI criteria. Total cholesterol (Tchol), high-density lipoprotein cholesterol (HDL-chol), and TG were assessed using enzymatic and colourimetric assays (Cobas Integra, Roche Diagnostics GmbH, Mannheim, Germany). The LDL-chol concentration was calculated using the Friedewald formula [17]. In patients with serum TG concentrations ≥ 400 mg/dL, LDL-chol was measured directly (BioSystems S.A., Barcelona, Spain). Non-HDL-chol was determined by subtracting HDL-chol from Tchol. Plasma eNOS, P, total calcium (Ca), parathyroid hormone (PTH), albumin, insulin, glycated haemoglobin (HbA_1c_), glucose, and other biochemical parameters were measured. After an overnight fast of a minimum of 8 h, venous blood was drawn into EDTA (or into heparin for the total calcium measurement) tubes and promptly centrifuged at 4 °C. The plasma was frozen at −80 °C in aliquots until analyses of the proteins were performed. The plasma eNOS levels were measured using the Human eNOS ELISA kit (cat. no. SEA868Hu, Wuhan USCN Business Co., Ltd., Wuhan, China) according to the manufacturer’s instructions. No significant cross-reactivity or interference was observed. The range was 0.156–10 ng/mL. The minimum detectable dose of eNOS was less than 0.057 ng/mL. The within-assay coefficient of variation (CV) was below 10%, and the CV for inter-assay precision was below 12%. Plasma phosphate levels were measured using the ammonium molybdate method, as a part of the multiple chemistry profile using the analyser Cobas Integra 400, Roche Diagnostics GmbH. The albumin-adjusted calcium was determined using the following equation in the setting of hypoalbuminemia: albumin-adjusted calcium (mg/dL) = (0.8 × (normal albumin (g/dL) − patient’s albumin)) + total calcium (mg/dL) [18]. The normal albumin was defaulted to 4 g/dL. The homeostasis model assessment of insulin resistance (HOMA–IR) was determined as fasting plasma insulin [µU/mL] × fasting plasma glucose (FPG) [mmol/L]/22.5. HbA_1c_ was determined using the turbidimetric inhibition immunoassay method (Cobas Integra; cat. no. 04528123; Roche Diagnostics GmbH). The concentration of high-sensitivity C-reactive protein (hsCRP) was determined using a high-sensitivity latex-enhanced immunoturbidimetry method (Cat. no. 04628918190; Cobas Integra 400, Roche Diagnostics GmbH). Other biochemical parameters were measured using standard laboratory techniques with a certified automated analyser (Cobas Integra 400, Roche, Basel, Switzerland).

### 2.6. Statistical Analysis

The normality of the distribution of variables was assessed separately for each group using the Shapiro–Wilk test. The data are presented as medians (interquartile ranges (Q1–Q3)) and means (SD) with the 95% confidence interval of means (95%CI). The categorical variables are presented as percentages. The Student’s *t*-test for unpaired data was used to compare groups with a normal distribution, and the Mann–Whitney U test was used if this condition was not fulfilled. Due to the lack of a normal distribution of plasma eNOS levels, Spearman’s rank correlation was performed between this variable and the other parameters in the groups. *p* < 0.05 was considered to indicate a statistically significant difference. Statistical analysis was performed using STATISTICA version 13 (TIBCO Software Inc., Palo Alto, CA, USA).

## 3. Results

### 3.1. Comparison between the Entire HD Group and the Control Group

The clinical and demographic characteristics of the patients on HD and the control patients are summarised in Table 1. Despite the prior declaration of good health, 57% of the subjects in the control group had dyslipidaemia, and a similar percentage was observed in the HD group (54%). The HD group had a percentage of cigarette smokers that was similar to that of the controls, and it had similar BMI, BP_sys_, PP, and MAP values to those of the controls. Compared with the control group, the HD group exhibited lower BP_dias_. In comparison with the controls, the patients on HD exhibited significantly elevated plasma P levels (5.4, 3.56–6.79 vs. 3.85, 2.96–4.43 mg/dL, respectively; *p* = 0.0027), Ca × P product, PTH, HOMA-IR, and serum TG levels. In turn, the entire HD group had lower plasma eNOS concentrations (0.725, 0.380–1.870 vs. 1.46, 1.20–1.69 ng/mL, respectively; *p* = 0.004), plasma Ca levels, RBCs, Tchol, HDL-chol, LDL-chol, and albumin levels (Table 2).

### 3.2. Comparison of Biochemical Parameters According to Plasma Phosphorus Tertiles in Patients on HD

The plasma levels of selected biochemical and clinical parameters in the different plasma P level tertiles are shown in Table 3. The HD patients showing P ≤ 3.4 mg/dL had a plasma eNOS concentration that was higher (3.75, 2.01–4.0 mg/mL) than that (0.570, 0.270–0.830 mg/mL) shown in the subjects with P ≥ 5.6 mg/dL (*p* = 0.0035) (Figure 2), whereas their median BP_sys_ was lower (120, 110–120 mmHg vs. 130, 120–140 mmHg, respectively; *p* = 0.04).

### 3.3. Comparison of Plasma Levels of eNOS, Phosphate, and Other Parameters between Patients on HD with and without D

The laboratory parameters of the patients on HD with and without D are presented in Table 4. There was no difference in the plasma P (*p* = 0.830), eNOS (*p* = 0.573), HgB, RBCs, hsCRP, HOMA-IR, HbA_1c_, serum HDL-chol, and albumin concentrations between the patients with and without D. In the patients on HD with D, the serum Tchol TG and LDL-chol levels were higher than those in the patients on HD without D. There was no difference in the dose of ESA per week or in the BP_sys_, BP_dias_, MAP, PP, or BMI between the patients with and without D (Table 5).

### 3.4. Comparison of Plasma Levels of eNOS, Phosphate, and Other Parameters in Controls with and without D

The control subjects with D did not differ significantly from those without D with respect to plasma eNOS (*p* = 0.477) and P (*p* = 0.175) levels.

### 3.5. Comparison of Plasma Levels of eNOS, Phosphate, and Other Parameters between Patients on HD with and without Type 2 DM

There was no difference in the plasma eNOS (*p* = 0.992), P (*p* = 0.648), HgB, RBCs, hsCRP, or albumin levels or in the lipid parameters between the patients with and without DM (Table 6). In the patients on HD with type 2 DM, the plasma HbA_1c_ levels and HOMA-IR were higher than those in the patients on HD without DM. There was no difference in the dose of ESA per week or in the BP_sys_, BP_dias_, MAP, PP, or BMI between the patients with and without DM (Table 5).

### 3.6. Correlation Analysis of eNOS Levels and Other Parameters in Patients on HD and Controls

In the entire HD group, there were negative correlations between the plasma eNOS levels and P (Figure 3) and the Ca × P product. There was no significant correlation between the plasma eNOS levels and the other laboratory, clinical, and anthropometric parameters in the entire HD group. In the HD group with D, there were negative correlations with BP_sys_ and BP_dias_. In the HD group without D, there were no significant correlations between the plasma eNOS and the other parameters. In the HD group with type 2 DM and without DM, there were no significant correlations between the plasma level of eNOS and the other examined parameters. In the control group, there were significant positive correlations between the plasma eNOS levels and P and Ca as well as the Ca × P product. The significant correlations of the plasma eNOS levels in the examined groups are presented in Table 7.

## 4. Discussion

To the best of our knowledge, this is the first clinical study to investigate the association between plasma P concentration and circulating eNOS levels in relation to other selected biochemical and clinical parameters in patients on HD. There are many studies on cell cultures which indicate that HP exposition significantly inhibits eNOS activity [5,8]. However, there are no studies that have examined the plasma circulating eNOS levels and their correlation with phosphataemia. In our preliminary study, we set out to investigate whether the relation observed between HP exposition and eNOS activity in the cell culture condition also exists in clinical settings. Because HP is a crucial clinical problem in patients with a decreased glomerular filtration rate [2,19,20], we conducted our study in patients with ESRD treated with intermittent HD. We showed that the circulating levels of eNOS in HD patients are markedly reduced in comparison with those of the healthy subjects. This observation is consistent with the results of studies on ESRD patients in which NO bioavailability was lower in comparison with that of the healthy population [2,3,4]. NO deficiency has been considered one of the important events leading to endothelial dysfunction in patients on HD [21]. The decrease in NO production prompted by an accumulation of endogenous eNOS inhibitors is an important cause of NO deficiency in these patients [7,21,22]. It is well known that HP is one of the uraemic toxins [4,5]. In our study, we showed that the plasma P levels in the patients on HD were significantly higher than those in the controls. In order to analyse the examined relationship more accurately, we divided the HD group into tertiles of P. The results of this analysis indicate that, as with the cell culture model, HP is associated with lower plasma eNOS concentrations. In the entire HD group, we found negative correlations between plasma eNOS levels and P as well as the Ca × P product. Surprisingly, in the controls, we found a positive correlation between plasma eNOS levels and P and the Ca × P product. As the plasma P levels were significantly lower in the controls in comparison with those of the HD group, this may suggest that plasma P levels within physiological ranges do not have an inhibitory effect on plasma eNOS levels and may even increase their levels. 

Such an explanation is consistent with the study results on cell culture by Peng A. et al. [8], in which hyper- and hypophosphataemia inhibited eNOS expression, while normophosphataemia did not change its expression.

Studies conducted by other researchers have shown that the plasma eNOS concentration strongly correlates with the tissue expression of this enzyme [23,24]. Thus, it seems that the measurement of the plasma eNOS concentration can be a much more accessible diagnostic method for assessing endothelial dysfunction than that of measuring it in a tissue sample.

Experimental studies in animal models have shown that diabetes mellitus and dyslipidaemia downregulate eNOS activity [25,26]. Thus, in the present study, our HD group was not only analysed as a whole group; it was also analysed following the division into groups of HD patients with type 2 DM and without DM and into groups of HD patients with D and without D. We showed that there was no difference in plasma eNOS concentrations between the HD patients with type 2 DM and without DM (despite the significant difference in the HbA_1c_ levels, *p* < 0.00001) or between the HD patients with D and without D.

It Is worth emphasising that in our study, there was no difference in plasma P levels between the groups with type 2 DM and those without DM or between the groups with D and without D. It seems that the accumulation of certain uraemic toxins that function as eNOS inhibitors, such as asymmetric dimethylarginine and advanced glycation end products (AGEs), were produced in similar concentrations in all of our HD patients, irrespective of the coexistence of diabetes mellitus or dyslipidaemia. These results are consistent with the observations of other researchers [27,28]. It is worth noting that our examined groups with type 2 DM and without DM as well as those with D and without D did not demonstrate differences in the examined clinical and demographic parameters, which could also have affected the lack of difference in the circulating eNOS levels. Many studies have revealed that patients on HD have many specific factors that impede eNOS activity, such as eNOS gene polymorphism [29], hyperparathyroidism [30], and oxidative stress [31]. The aforementioned uraemic eNOS inhibitors can dim and obliterate the effects of diabetes mellitus and dyslipidaemia on eNOS activity in HD patients. Previous reports showed a decline in the NO–eNOS axis bioactivity during HD sessions [4,5]. However, the mechanisms of these alterations in HD patients are still not fully understood.

In HD patients, elevated, poorly controlled blood pressure is common and impairs the function of endothelial cells in general [32]. Understanding the mechanisms and evaluating and defining the best management of blood pressure in patients on HD is a significant challenge for clinical researchers. Although volume overload and sodium retention appear to be the main pathogenic mechanisms of hypertension in this population, other factors, such as oxidative stress, the use of recombinant erythropoietin, and the uncoupling of eNOS, may also be involved [24,33,34]. In the present study, we showed that in the patients on HD with D, the plasma eNOS concentration correlated negatively with BP_sys_ and BP_dias_. Based on the data obtained from our preliminary study, these findings may not reliably explain why these correlations were observed only in D patients. However, many studies on cell culture indicate such a relationship [19,20]. Cosentino et al. revealed in an animal model that hypertensive rats produce superoxide inhibitors of tetrahydrobiopterin (BH_4_), which is an essential cofactor of eNOS [35]. Insufficiency of BH_4_ leads to the uncoupling of eNOS and lowers eNOS activity [36,37,38]. Therefore, it can be assumed that the negative association between BP_sys_ and BP_dias_ and the plasma eNOS levels observed in our study may be related to the aforementioned phenomenon. This relation was more clearly evident in the group of patients with D and may result from the fact that 94% of these patients were treated with atorvastatin for 12 weeks before the start of the study. Antoniades et al. showed that in patients with normal renal function treated with atorvastatin, eNOS recoupling and elevated vascular BH_4_ availability were observed [39]. In our opinion, these observations may suggest the potential positive effect of previous statin treatment on plasma eNOS levels. The interval of treatment with statin, and the subsequent greater sensitivity of eNOS–NO, could result in an inhibitory effect of systolic and diastolic blood pressure on the plasma eNOS concentration.

Some of the limitations of this study should be acknowledged. First, a cross-sectional design diminishes the ability to establish a causal relationship. Second, all the patients in our study were Caucasian, and potential differences with other ethnicities were not examined. Third, the number of patients and healthy subjects was relatively small. Nevertheless, the examined HD population had exposed vascular abnormalities and ongoing and constant clinical observation, which allowed a better definition of the exclusion criteria and the precise matching of the groups with the inclusion criteria. Larger studies and more diversified groups are warranted to establish whether these relationships are indeed linked in the HD population.

In conclusion, the plasma levels of eNOS in the HD patients were reduced in comparison with those in the healthy subjects. HP and high blood pressure appeared to decrease the circulating eNOS levels. These findings demonstrate an additional negative impact of HP on eNOS activity. Further studies will shed more light on these observations.

## Figures and Tables

**Figure 1 biomedicines-12-00687-f001:**
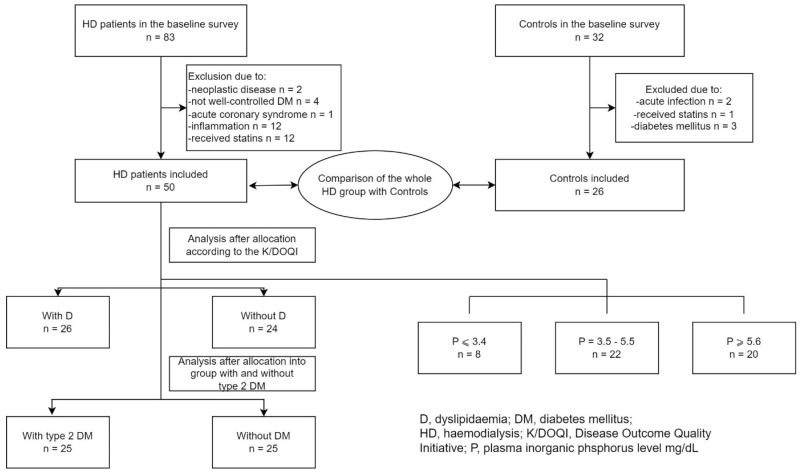
The flow diagram of the study.

**Figure 2 biomedicines-12-00687-f002:**
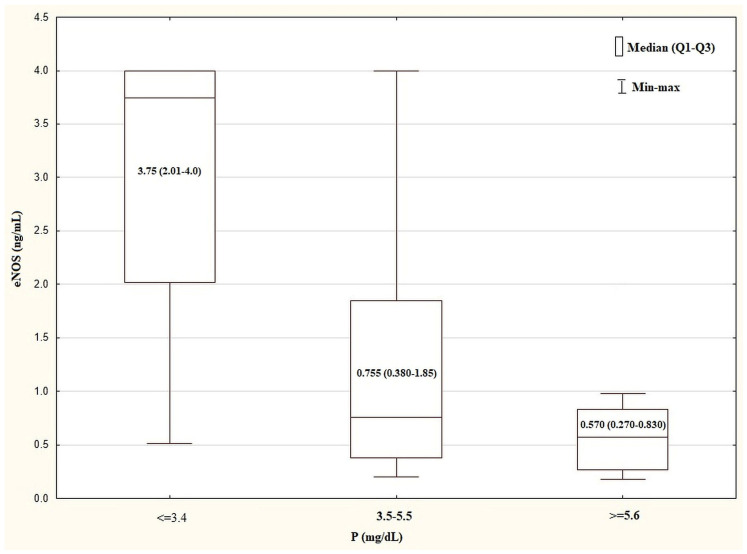
Comparison of plasma eNOS concentration in the tertiles of plasma P levels in HD group. Kurskal–Wallis test: *p* = 0.0035. The post test for trend: *p* = 0.0008. eNOS, endothelial nitric oxide synthase; HD, haemodialysis; P, inorganic phosphorus; Q—quartile.

**Figure 3 biomedicines-12-00687-f003:**
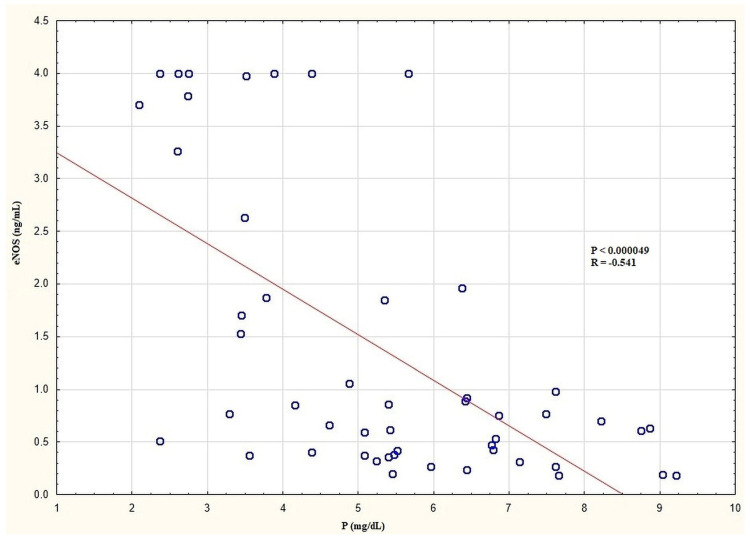
Correlation between plasma eNOS and P levels in the entire HD group. eNOS, endothelial nitric oxide synthase; HD, haemodialysis; P, inorganic phosphorus.

**Table 1 biomedicines-12-00687-t001:** Demographic parameters and clinical data of the entire group of haemodialysis patients and controls.

Parameter	Controls n = 26 Median (Q1–Q3)	HD Patients n = 50 Median (Q1–Q3)	*p* Value
Age (years)	69 (64–76)	66 (60–74)	0.061
Gender, M/F	11/15	27/23	-
Dyslipidaemia, n (% of all)	15 (57)	26 (54)	0.693
Cigarette smoking, n (% of all)	1 (4)	4 (8)	0.837
Body mass index, kg/m^2^	27.2 (24.1–30.1)	26.9 (23.6–29.2)	0.660
ESA, IU/kg body weight/per week	-	30.2 (12–104)	-
Calcium-containing phosphate binders, n (%)	-	38 (76)	-
Non-calcium-containing phosphate binders, n (%)	0	0	-
Alfacalcidol, n (%)	0	24 (48)	-
Cinacalcet, n (%)	0	0	-
BP_sys._, mm Hg	130 (120–140)	120 (100–164)	0.078
BP_dias._, mm Hg	80 (75–90)	70 (70–80)	0.002
PP, mm Hg	50 (45–55)	50 (40–60)	0.715
MAP, mm Hg	66.7 (60–73.3)	66.7 (77.7–67.1)	0.714
Basal disease, n (% of subjects)			
Diabetic nephropathy	0	25 (50)	-
Chronic glomerulonephritis	0	12 (24)	-
Polycystic kidney disease	0	5 (10)	-
Chronic tubulointerstitial nephritis	0	3 (6)	-
Other/unknown	0	5 (10)	-

Abbreviations: BP_sys_—systolic blood pressure; BP_dias_—diastolic blood pressure; ESA—erythropoiesis stimulating agent; F—female; HD—haemodialysis; M—male; MAP—mean arterial pressure; PP—pulse pressure; Q—quartile.

**Table 2 biomedicines-12-00687-t002:** Values of laboratory parameters in haemodialysis patients and controls.

Parameter	Controls n = 26 Median (Q1–Q3)	HD Patients n = 50 Median (Q1–Q3)	*p* Value
eNOS, ng/mL	1.46 (1.2–1.69)	0.725 (0.380–1.870)	0.004
P, mg/dL	3.85 (2.96–4.43)	5.4 (3.56–6.79)	0.0027
Ca, mg/dL	9.18 (8.92–9.74)	8.43 (7.98–8.99)	0.00001
Ca × P	35.1 (26.7–41.4)	45.2 (33.8–56.7)	0.026
PTH, pg/mL	27.4 (22.1–47.4)	483 (324–740)	<0.0001
HgB, g/dL	13.6 (12.8–14.3	11.6 (10.3–12.0)	0.070
RBC	4.6 (4.2–4.8)	3.81 (3.42–3.96)	<0.000001
Tchol, mg/dL	195.5 (178–234)	170 (137–196)	0.001
LDL-chol, mg/dL	102.5 (89–139)	91.5 (65–117)	0.005
HDL-chol, mg/dL	64.5 (52.0–79.0)	38.0 (33.0–49.0)	0.005
TG, mg/dL	90 (72–112)	161 (108–206)	<0.0001
hsCRP, mg/L	4.8 (2.1–5.1)	8.1 (4.0–18.4)	0.050
HOMA	2.31 (1.6–2.92)	6.34 (2.62–9.52)	0.0002
HbA_1c_	5.4 (5.2–5.6)	5.75 (5.0–7.0)	0.130
Albumin, g/dL	4.4 (4.2–4.6)	3.7 (2.9–4.1)	<0.000001

Abbreviations: Ca—calcium; eNOS—endothelial nitric oxide synthase; HbA_1c_—glycated haemoglobin; HD—haemodialysis; HDL-chol—high-density lipoprotein cholesterol; HgB—haemoglobin; hsCRP—high-sensitivity C reactive protein; HOMA—homeostatic model assessment; LDL—low-density lipoprotein cholesterol; P—phosphate; PTH—parathyroid hormone; RBC—red blood cells; Tchol—total cholesterol; TG—triglycerides; Q—quartile.

**Table 3 biomedicines-12-00687-t003:** Biochemical and clinical parameters according to plasma phosphorus tertiles in patients on haemodialysis.

		HD Patients	n = 50	
	T_1_ (n = 8)	T_2_ (n = 22)	T_3_ (n = 20)	*p* for Trend
Phosphorus, mg/dL	≤3.4 Median (Q1–Q3)	3.5–5.5 Median (Q1–Q3)	≥5.6 Median (Q1–Q3)	
eNOS, ng/mL	3.75 (2.01–4.0)	0.755 (0.380–1.85)	0.570 (0.270–0.830)	0.0008
HgB, g/dL	11.4 (9.8–11.9)	11.6 (10.2–12.2)	11.8 (11.1–12.5)	0.091
RBC, 10^6^/µL	3.82 (3.62–4.05)	3.73 (3.33–3.87)	3.86 (3.58–3.96)	0.136
P, mg/dL	2.6 (2.37–2.74)	4.74 (3.78–5.4)	7.0 (6.44–7.94)	<0.000001
Ca, mg/dL	8.43 (8.31–8.83)	8.82 (8.06–9.03)	8.31 (7.89–8.98)	0.218
Ca × P, mg^2^/dL^2^	22.2 (18.8–24.4)	40.6 (33.8–44.6)	58.3 (53.8–66.6)	0.200
PTH, pg/mL	538 (204–972)	465 (252–662)	500 (414–730)	0.267
T_chol_, mg/dL	159 (128–195)	174 (146–206)	168 (139–191)	0.395
LDL-chol, mg/dL	85 (62–114)	94.5 (76–117)	92.5 (62.5–119)	0.273
HDL-chol, mg/dL	40 (33–57)	38.5 (34–45)	36.5 (31.5–50)	0.252
TG, mg/dL	134 (99–200)	162 (99–235)	161 (115–224)	0.285
hsCRP, mg/L	13.2 (6.75–23.6)	6.9 (4.0–18.4)	4.7 (4.0–20.7)	0.301
HOMA	3.37 (0.976–16.7)	6.73 (3.19–9.5)	5.61 (3.17–9.97)	0.392
HbA_1c_, %	5.6 (4.9–7.3)	6.3 (5.3–7.2)	5.4 (5.0–7.6)	0.304
Age, years	74 (67–79)	65 (53–67)	65.5 (61–74)	0.307
BMI, kg/m^2^	24.6 (20.6–29.4)	27.8 (24.8–29.2)	25.5 (23.8–29.9)	0.485
HD vintage, months	50.4 (35.7–66.6)	29.7 (15.1–47.4)	22.9 (13–41.9)	0.211
BP_sys_, mmHg	120 (110–120)	125 (110–130)	130 (120–140)	0.021
BP_dias_, mmHg	70 (70–75)	70 (70–80)	80 (70–80)	0.317
PP	50 (40–50)	50 (40–60)	50 (40–60)	0.091
MAP	66.7 (53.3–66.7)	66.7 (53.3–80)	66.7 (53.3–80)	0.370

Abbreviations: BMI—body mass index; BP_dias_—diastolic blood pressure; BP_sys_—systolic blood pressure; Ca—calcium; eNOS—endothelial nitric oxide synthase; HbA_1c_—glycated haemoglobin; HD—haemodialysis; HDL-chol—high-density lipoprotein cholesterol; HgB—haemoglobin; hsCRP—high-sensitivity C reactive protein; HOMA—homeostatic model assessment; LDL—low-density lipoprotein cholesterol; MAP—mean arterial pressure; P—phosphate; PP—pulse pressure; PTH—parathyroid hormone; RBC—red blood cells; T—tertile; T_chol_—total cholesterol; TG—triglycerides; Q—quartile.

**Table 4 biomedicines-12-00687-t004:** Values of laboratory parameters in haemodialysis patients with dyslipidaemia and without dyslipidaemia.

Parameter	HD with D n = 26 Median (Q1–Q3)	HD without D n = 24 Median (Q1–Q3)	*p* Value
eNOS, ng/mL	0.725 (0.400–2.63)	0.700 (0.375–1.615)	0.573
P, mg/dL	5.16 (4.16–6.86)	5.43 (3.44–6.44)	0.830
Ca, mg/dL	8.38 (8.06–8.85)	8.70 (7.89–9.07)	0.534
Ca × P	43.9 (35.4–56.7)	47.0 (29.3–56.9)	0.808
PTH, pg/mL	465 (275–740)	502 (379–797)	0.534
HgB, g/dL	11.6 (10.2–12.4)	11.8 (10.5–12.3)	0.690
RBC, 10^6^/µL	3.80 (3.33–3.98)	3.81 (3.57–3.93)	0.688
T_chol_, mg/dL	194 (180–231)	136 (125–154)	<0.0001
LDL-cholesterol, mg/dL	116 (101–135)	64.0 (52.5–79.0)	<0.0001
HDL-cholesterol, mg/dL	38.0 (34.0–49.0)	38.5 (31.5–54.0)	0.953
TG, mg/dL	204 (148–289)	111 (87.0–162)	<0.0001
hsCRP, mg/L	4.7 (4.0–18.0)	9.45 (4.0–24.0)	0.410
HOMA	7.11 (2.99–9.52)	5.55 (2.13–9.88)	0.478
HbA_1c_, %	6.1 (5.2–7.5)	5.7 (5.0–7.2)	0.748
Albumin, g/dL	3.75 (3.5–4.0)	3.6 (3.4–3.8)	0.058

Abbreviations: Ca—calcium; D—dyslipidaemia; eNOS—endothelial nitric oxide synthase; HbA_1c_—glycated haemoglobin; HD—haemodialysis; HDL-_chol_—high-density lipoprotein cholesterol; HgB—haemoglobin; hsCRP—high-sensitivity C reactive protein; HOMA—homeostatic model assessment; LDL—low-density lipoprotein cholesterol; P—phosphate; PTH—parathyroid hormone; RBC—red blood cells; T_chol_—total cholesterol; TG—triglycerides; Q—quartile.

**Table 5 biomedicines-12-00687-t005:** Demographic parameters and clinical data of haemodialysis patients with and without dyslipidaemia as well as with and without type 2 diabetes mellitus.

	With D Median (Q1–Q3)	Without D Median (Q1–Q3)	*p* Value	With Type 2 DM Median (Q1–Q3)	Without DM Median (Q1–Q3)	*p* Value
N subject	26	24	-	25	25	-
Age (years)	64.5 (60.0–72.0)	66.5 (60.0–74.5)	0.938	67.0 (59.0–73.0)	65.0 (61.0–74.0)	0.938
Gender (M/F)	12/14	15/9	–	14/11	13/12	-
HD vintage, months	31.3 (15.4–59.8)	25.9 (11.1–46.3)	0.084	28.3 (15.1–46.0)	31.6 (15.3–67.7)	0.097
BMI (kg/m^2^)	26.9 (23.9–29.1)	27.0 (20.9–29.6)	0.742	28.4 (24.8–30.4)	25.1 (23.5–28.0)	0.235
BP_sys_ (mmHg)	120 (110–140)	125 (115–135)	0.588	120 (120–140)	120 (110–140)	0.169
BP_dias_ (mmHg)	70 (70–80)	70 (70–80)	0.824	70 (70–80)	70 (70–80)	0.651
PP	50 (40–60)	50 (40–60)	0.725	50 (50–60)	50 (40–60)	0.133
MAP	66.6 (53.3–80.0)	66.6 (53.3–80.0)	0.726	66.6 (66.6–80.0)	66.6 (52.5–80.0)	0.132

Abbreviations: BMI, body mass index; BP_dias_, diastolic blood pressure; BP_sys_, systolic blood pressure; DM, diabetes mellitus; D, dyslipidaemia; F—female; M—male; MAP, mean arterial pressure; N, number; PP, pulse pressure; Q—quartile.

**Table 6 biomedicines-12-00687-t006:** Values of laboratory parameters in haemodialysis patients with type 2 diabetes mellitus and without diabetes mellitus.

Parameter	HD with Type 2 DMn = 25 Median (Q1–Q3)	HD without DM n = 25 Median (Q1–Q3)	*p* Value
eNOS, ng/mL	0.660 (0.420–1.70)	0.750 (0.380–1.96)	0.992
P, mg/dL	5.0 (3.56–6.44)	5.4 (3.8–6.8)	0.648
Ca, mg/dL	8.34 (7.88–9.03)	8.81 (8.26–8.95)	0.409
Ca × P	42.3 (33.7–54.1)	46.3 (35.4–58.0)	0.497
PTH, pg/mL	466 (283–662)	515 (420–815)	0.460
HgB, g/dL	11.6 (10.3–12.2)	11.6 (10.9–12.4)	0.649
RBC, 10^6^/µL	3.8 (3.4–4.1)	3.82 (3.46–3.91)	0.586
Tchol, mg/dL	166 (146–192)	178 (135–196)	0.763
LDL-cholesterol, mg/dL	89.0 (63.0–104)	100 (73.0–117)	0.210
HDL-cholesterol, mg/dL	39.0 (33.0–49.0)	38.0 (34.0–47.0)	0.719
TG, mg/dL	162 (114–235)	161 (92.0–202)	0.547
hsCRP, mg/L	8.7 (4.0–23.3)	4.0 (4.0–18.3)	0.240
HOMA	9.5 (6.5–24.1)	2.99 (1.11–4.34)	<0.0001
HbA_1c_, %	7.2 (6.4–8.0)	5.0 (4.8–5.3)	<0.00001
Albumin, g/dL	3.7 (3.4–3.8)	3.7 (3.5–3.9)	0.703

Abbreviations: Ca—calcium; DM—diabetes mellitus; eNOS—endothelial nitric oxide synthase; HbA_1c_—glycated haemoglobin; HD—haemodialysis; HDL-chol—high-density lipoprotein cholesterol; HgB—haemoglobin; hsCRP—high-sensitivity C reactive protein; HOMA—homeostatic model assessment; LDL—low-density lipoprotein cholesterol; P—phosphate; PTH—parathyroid hormone; RBC—red blood cells; Tchol—total cholesterol; TG—triglycerides; Q—quartile.

**Table 7 biomedicines-12-00687-t007:** Significant correlations between endothelial nitric oxide synthase and other biochemical parameters in the examined groups treated with haemodialysis and controls.

Groups and Correlated Parameters	R	*p* Value
**Whole HD group, n = 50**		
eNOS and P	−0.541	<0.000049 ^b^
eNOS and Ca × P	−0.542	0.000049 ^b^
**Controls, n = 26**		
eNOS and P	0.373	0.059
eNOS and Ca	0.469	0.015 ^a^
eNOS and Ca × P	0.424	0.030 ^a^
**Dyslipidemic HD group, n = 26**		
eNOS and BP_sys_	−0.401	0.042 ^a^
eNOS and BP_dias_	−0.440	0.021 ^a^

Abbreviations: ^a^
*p* < 0.05; ^b^
*p* < 0.001; BP_dias_—diastolic blood pressure; BP_sys_—systolic blood pressure; Ca—calcium; eNOS—endothelial nitric oxide synthase; HD—haemodialysis; P—phosphorus.

## Data Availability

Data are contained within the article.

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
