# Peer review of "Association of Circulating Endothelial Nitric Oxide Synthase Levels with Phosphataemia in Patients on Haemodialysis"

_biomedicines, 2024, doi:10.3390/biomedicines12030687_

Round 1
Reviewer 1 Report
Comments and Suggestions for Authors
Hyperphosphatemia (HF) can induce endothelial damage and significantly disrupt eNOS expression. The authors evaluated the relationship between plasma inorganic phosphorus (P) levels and circulating plasma eNOS concentration in hemodialysis (HD) patients. Positive correlations were found between circulating eNOS levels and plasma P concentrations. In HD patients with D, higher systolic and diastolic blood pressures were accompanied by reduced plasma eNOS concentrations. The topic is inovative and interesting from clinical viewpoint. No remarks in framing the thesis, materials and method , results. The discussion is detailed. I suggest changing the conclusion part and adding references from the last 5 years, if possible.
Comments on the Quality of English Language
Minor editing of English language required
Reviewer 2 Report
Comments and Suggestions for Authors
The main goal of the present was to know if there is any relation between circulating levels of inorganic phosphorous (P) and of those of eNOS in patients with hemodialysis (HD). Comparisons between P and eNOS levels were performed in 50 HD patients and 26 control subjects. Additional comparisons were performed in HD patients divided according to the presence of dyslipidemia and or type 2 diabetes mellitus. Also, a correlation of eNOS levels with P was performed in HDP patients according to three values of P: <3.4 (n=8), 3.5 – 3.5 (n=22) and >5.6 mg/dL (n= 20). eNOS levels were lower in HD patients and eNOS levels correlated negatively with P levels in these patients. The corresponding ethical Committee approved the study. This is an original study. The information presented in this study in HD patients is new and support data of the literature of previous in vitro studies. In addition the authors describe the limitations of the study. The conclusions are supported by the data presented and are related to the main question of the study
Major
The eNOS values in the HD group were 0.380 – 1.870 ng/ml (line 183). In contrast eNOS values presented in Figure 2 were 2.01 – 4.0 ng/ml (for P values less or equal to 3.4 mg/dL), 0.380 – 1.85 ng/ml (for P values from 3.5 – 5.5 mg/dL) and 0.270-0830 ng/ml (for P values higher or equal to 5.6 mg/dL). It is not clear for this reviewer why the values are different (between those presented in line 183 and in Figure 2) if the patients were the same.
The methods should be described in more detail and references for each biochemical determination should be given.
Questions
The measured the eNOs protein levels. What about eNOS activity?, NO levels and GFR were measured?
What is the postulated mechanism by which high P levels decreased eNOs levels? eNOS transcription or synthesis are altered?
Minor points
Abstract
“Positive correlations were found between circulating levels of eNOS and plasma P concentrations.” It must be stated the group to which is referred
Quality of Figure 1 must be increased. Letters are small and blurred
Abstract: eNOS and ESDR should be defined.
Line 45: “ECs” should be defined.
The location (city and country) of all the providers should be given the first time they are mentioned.
Tables 1 and 2. It is suggested that the data of control subjects be located before the data of HD patients (column 3 should be first that column 2).
Line 346. References 38 and 38 are not from Cosentino et al. Ref 38 is of Higashi et al and ref 39 is of Schmidt et al.
Line 354. Reference 40 is not form Charalambos et al. Instead of it is of Antoniades et al.
The following abbreviations are not in the list of abbreviations:
M/F, ESC/EAS, ELISA, EDTA, CV, CI, and FPG.
All the abbreviations used in tables or figures must be defined below.
Author Response
Please see the attachement.
